# Antiviral Activities of Green Tea Components against Grouper Iridovirus Infection In Vitro and In Vivo

**DOI:** 10.3390/v14061227

**Published:** 2022-06-05

**Authors:** Pengfei Li, Shuaishuai Huang, Shuangyan Xiao, Youhou Xu, Xinxian Wei, Jun Xiao, Zhongbao Guo, Qing Yu, Mingzhu Liu

**Affiliations:** 1Guangxi Key Laboratory of Beibu Gulf Marine Biodiversity Conservation, College of Marine Sciences, Beibu Gulf University, Qinzhou 535000, China; pfli2016@gxas.cn (P.L.); 13797226983@163.com (S.H.); xyh_2016@bbgu.edu.cn (Y.X.); 2Guangxi Key Laboratory of Aquatic Biotechnology and Modern Ecological Aquaculture, Guangxi Engineering Research Center for Fishery Major Diseases Control and Efficient Healthy Breeding Industrial Technology (GERCFT), Guangxi Academy of Sciences, Nanning 530015, China; 15737954163@163.com; 3Guangxi Key Laboratory of Aquatic Genetic Breeding and Healthy Breeding, Guangxi Academy of Fishery Science, Nanning 530015, China; Byang15@126.com (X.W.); dreamshaw@foxmail.com (J.X.); guozhongbaono1@163.com (Z.G.)

**Keywords:** green tea component, antiviral activity, grouper iridovirus, tea polyphenol, EGCG

## Abstract

(1) Background: Singapore grouper iridovirus (SGIV) can cause extensive fish deaths. Therefore, developing treatments to combat virulent SGIV is of great economic importance to address this challenge to the grouper aquaculture industry. Green tea is an important medicinal and edible plant throughout the world. In this study, we evaluated the use of green tea components against SGIV infection. (2) Methods: The safe working concentrations of green tea components were identified by cell viability detection and light microscopy. Additionally, the antiviral activity of each green tea component against SGIV infection was determined with light microscopy, an aptamer (Q5c)-based fluorescent molecular probe, and reverse transcription quantitative PCR. (3) Results: The safe working concentrations of green tea components were green tea aqueous extract (GTAE) ≤ 100 μg/mL, green tea polyphenols (TP) ≤ 10 μg/mL, epigallocatechin-3-gallate (EGCG) ≤ 12 μg/mL, (-)-epigallocatechin (EGC) ≤ 10 μg/mL, (-)-epicatechin gallate (EGC) ≤ 5 μg/mL, and (-)-epicatechin (EC) ≤ 50 μg/mL. The relative antiviral activities of the green tea components determined in terms of *MCP* gene expression were TP > EGCG > GTAE > ECG > EGC > EC, with inhibition rates of 99.34%, 98.31%, 98.23%, 88.62%, 73.80%, and 44.31%, respectively. The antiviral effect of aptamer-Q5c was consistent with the results of qPCR. Also, TP had an excellent antiviral effect in vitro, wherein the mortality of fish in only the SGIV-injection group and TP + SGIV-injection group were 100% and 11.67%, respectively. (4) Conclusions: In conclusion, our results suggest that green tea components have effective antiviral properties against SGIV and may be candidate agents for the effective treatment and control of SGIV infections in grouper aquaculture.

## 1. Introduction

Groupers are fish in the genus *Epinephelus* and the family Serranidae [1]. This high-quality edible fish has a low fat content and high protein content. Groupers are highly sought after for their meat quality, their delicious flavor, and tenderness. Given their high cost and economic value, groupers have become an important product in marine aquaculture in recent years. According to the literature, most grouper species in China are distributed in the South China Sea. Breakthroughs in the regulation of reproductive growth and artificial breeding technologies have gradually transformed grouper farming in China into a large-scale aquaculture industry. However, with the expansion of caged aquaculture, the various diseases affecting grouper populations in southern China have become increasingly serious [2,3,4]. Singapore grouper iridovirus (SGIV) is a recently discovered serious infectious virus of fish and is currently the main pathogen causing severe economic losses in grouper aquaculture. The viral particles are about 200 nm in diameter, with a hexagonal profile, and can be observed in the cytoplasm of infected cells [5]. Grouper-iridovirus-infected groupers show typical clinical signs of inappetence, enlargement of the liver and spleen, and large-scale mortality within several days. An effective method of controlling SGIV infections is urgently required [6,7]. Given that it is a highly pathogenic virus, SGIV can cause extensive fish deaths in a short period. Therefore, treatments that inhibit the replication of SGIV are extremely economically important to address this challenge to the grouper aquaculture industry.

China has an extensive history of using medicinal plants to prevent and treat viral diseases. Such treatments are natural, efficient, and inexpensive. They also have low toxicity, cause minimal adverse effects, and do not usually induce the development of drug resistance. In herbal remedies, the medicinal material itself contains alkaloids, glycosides, carbohydrates, organic acids, volatile oils, flavonoids, tannins, resins, oils, pigments, glycopeptides, amino acids, and a variety of trace elements. These components act both independently and synergistically. Such natural compounds also supplement animal nutrition, promote growth, and exert antibacterial and immune effects. In recent years, herbal remedies have become more frequently used in aquaculture [8,9]. Green tea is an important homologous medicine and food in China. The health benefits of unfermented green tea are largely attributed to the catechins in it [10], such as tea polyphenols (TP), (-)-epigallocatechin gallate (EGCG), (-)-epigallocatechin (EGC), (-)-epicatechin gallate (ECG), (-)-epicatechin (EC), and (-)-catechin (C) [11]—the structures of which are presented in Figure 1. The contributions of green tea and its major constituent polyphenols to bodily health are well known, including their antiviral, antioxidative, and antimicrobial activities [12,13]. Several studies have shown that EGCG inhibits grass carp reovirus infection in a *Ctenopharyngodon idellus* kidney cell line [14].

In this study, we tested the antiviral effects of a series of green tea components, including green tea aqueous extracts (GTAE), TP, EGCG, ECG, EGC, and EC, against SGIV. Of these components, GTAE, TP, and EGCG showed remarkable anti-SGIV activity, with inhibitory rates of >99%. Our in vivo anti-SGIV results also suggest that TP reduces grouper mortality.

## 2. Materials and Methods

### 2.1. Fish, Cells, Virus, and Reagents

Hybrid groupers (*Epinephelus fuscoguttatus*♀ × *E. lanceolatus*♂) with a mean body length of 6.10 ± 0.37 cm were cultured in seawater (30‰ salinity) at 28 °C. Grouper spleen (GS) cells were cultured in Leibovitz’s L15 medium (Gibco, Grand Island, NY, USA) supplemented with 10% fetal bovine serum (Gibco) at 28 °C (Qin, et al., 2006). The virus was isolated from diseased hybrid grouper (*E. fuscoguttatus*♀ × *E. lanceolatus*♂) in Guangxi Province and maintained in our laboratory [15]. SGIV was propagated in GS cells at 28 °C. The viral titers, expressed as the 50% tissue culture infective dose (TCID_50_), reached 106.5 TCID_50_/mL. The active ingredients of green tea used in this study included TP, EGCG, ECG, EGC, and EC, which were isolated from green tea (>98% purity) and purchased from Macklin Co., Ltd. Each compound was dissolved in dimethyl sulfoxide (DMSO; Sigma, Germany) and stored at −20 °C. The stock solution was diluted to the working concentrations in L15 medium (pH 7.5). Diluted DMSO in L15 medium was used as the control.

### 2.2. Preparation of GTAE

Green tea was purchased from Yixin Pharmacy (Nanning, Guangxi, China) and ground to powder. The powder (50 g) was macerated in water, incubated for 12 h at 4 °C, and then boiled for 4 h until the concentration of GTAE was 250 mg/mL, as described previously [8]. The GTAE was then centrifuged for 10 min at 5000× *g* and 4 °C, and the supernatant was collected and filtered through a sterile nylon net filter with a 0.22 μm pore size on the bench top.

### 2.3. Analysis of the Safe Working Concentrations of Green Tea Components with a Cell Viability Assay

The effects of the different concentrations of the green tea components on cell viability were determined as described previously [16]. GS cells were cultured at a density of 1 × 10^6^ in a 96-well plate (Corning, Shanghai, China) for 18 h at 28 °C. The cells were then incubated with different concentrations of green tea components for 48 h at 28 °C. In the control group, GS cells were incubated with diluted DMSO in L15 medium. After incubation for 48 h at 28 °C, the cells were examined with light microscopy to detect signs of cytotoxicity. The cells were then washed twice with phosphate-buffered saline (PBS) and incubated with CCK-8 solution (Beyotime, Shanghai, China) diluted to 10% with PBS for 4 h at room temperature. Cell viability was measured as the absorbance of the cells at 450 nm in an ELISA plate reader (Thermo Fischer Scientific, Waltham, MA, USA). The experiment was performed three times in quadruplicate and the results are presented as means ± standard deviations (SD).

### 2.4. Total RNA Extraction and Reverse Transcription (RT) Quantitative PCR (qPCR) Analysis

The cells and culture supernatants from each well were collected for total RNA extraction. The total RNA was extracted with TRIzol Reagent (Cwbio, Nanning, China) and reverse transcribed into cDNA with a reverse transcription kit (Cwbio, Nanning, China). SGIV infection was detected in the GS cells with RT qPCR targeting the SGIV major capsid protein gene (*MCP*) and viral envelope protein gene (*VP19*), as previously described [1]. The grouper β-actin gene was used as the internal control [17]. The primer pairs used are listed in Table 1. qPCR amplification was performed with the thermal cycling conditions: 95 °C for 10 min, 40 cycles of 95 °C for 15 s, and 60 °C for 1 min. A melting curve analysis was used to verify the specificity of the amplified products. Each individual sample was run in triplicate. The qPCR data were analyzed with the 2^−^^ΔΔCt^ method and are shown as means ± SD.

### 2.5. Monitoring SGIV Infection with an Aptamer (Q5c)-Based Fluorescent Molecular Probe (Q5c-AFMP)

Aptamer Q5c was successfully selected against SGIV-infected fish cells in a previous study [2] and is an extremely sensitive molecular probe for SGIV. The sequence of aptamer Q5c is 5′-TATTCGGGTTATTGCTCCTCTTTATTGTCACCTGGATGTATGATCG-TGTAG-3′. Aptamer Q5c was synthesized by Sangon Biotech (Shanghai, China) and labeled with 6-carboxy-fluorescein (Q5c-AFMP). In this study, Q5c-AFMP was used to detect SGIV infection with flow cytometry. The cells and culture supernatants were collected at 48 h postinfection (hpi) and incubated with Q5c-AFMP (300 nM) for 30 min at 4 °C. The cells were washed three times with PBS and resuspended in 200 μL of PBS for flow-cytometric analysis. The means ± SD of the results of three independent experiments were calculated.

### 2.6. Inhibitory Percentage of Green Tea Components against SGIV Infection

The RT qPCR was used to evaluate the inhibitory percentage of green tea components against SGIV infection. The inhibitory percentage equation was calculated by: The inhibitory percentage (%) = 100 − (Test-Control)/(SGIV-Control) × 100%; which, test represents the RT-qPCR results of green tea components group treated with SGIV; con represents RT-qPCR results of cells without treatments; SGIV represents the RT qPCR results of group treated with SGIV only.

### 2.7. Antiviral Activity of Green Tea Components against SGIV Infection In Vitro

The antiviral activity of each green tea component against SGIV infection in vitro was assessed as described previously [18]. GS cells were cultured at a density of 1 × 10^6^ in a 12-well plate (Corning) for 18 h at 28 °C. The green tea components were diluted to safe working concentrations with L15 medium and were added with SGIV (multiplicity of infection [MOI] = 1) to the GS cells and incubated for 48 h at 28 °C. The negative control group (Con1) contained GS cells incubated with DMSO diluted in L15 medium.

The positive control (Con2) contained GS cells incubated with DMSO diluted in L15 medium and SGIV only (MOI = 1). At 48 hpi, the cells and culture supernatants were collected for total RNA extraction and RT qPCR analysis. Q5c-AFMP was also used to confirm the inhibitory effects of green tea components on SGIV infection. The means ± SD of three independent experiments were calculated.

### 2.8. Antiviral Activity of Green Tea Components against SGIV Infection In Vivo

Hybrid groupers (*E. fuscoguttatus*♀ × *E. lanceolatus*♂), approximately 6 cm in body length, were purchased from a fish farm in Beihai city, Guangxi, China. Before the experiment, the groupers were acclimated for seven days. The 240 hybrid groupers were divided into four groups in triplicate: (i) the PBS-treated negative control; (ii) treated with SGIV only; (iii) treated with green tea component only; and (iv) treated with SGIV and green tea component. Each grouper was injected intraperitoneally with a 50 μL treatment volume. Total RNA was extracted from the livers, spleens, and kidneys of selected fish at 24, 48, and 72 h. Mortality was recorded daily for 10 days. The survival rate of each group was calculated by counting the numbers of surviving groupers. The results were calculated as means ± SD.

### 2.9. Statistical Analysis

The average values of three independent experiments were calculated. Intergroup differences were compared with a one-way analysis of variance with SPSS software. *p* < 0.05 was considered to indicate statistically significant differences.

## 3. Results

### 3.1. Safe Working Concentrations of Green Tea Components

Morphological changes were observed in the GS cells incubated for 48 h with the green tea components. Normal GS cells were used as the control group, which were incubated with DMSO diluted in L15 medium to maintain normal cell growth (Figure 2A). Cell survival was >99% (Figure 2B). When GS cells were incubated with GTAE ≥ 200 μg/mL, TP ≥ 20 μg/mL, EGCG ≥ 24 μg/mL, or EGC ≥ 20 μg/mL, or the cell morphologies changed significantly—showing cell rounding, shrinking, and detachment (Figure 2A)—and the cell survival rate differed significantly from that of the control group (Figure 2B). No cytotoxic effects were observed, even with 50 μg/mL EC. When the concentrations of the green tea components were GTAE ≤ 100 μg/mL, TP ≤ 10 μg/mL, EGCG ≤ 12 μg/mL, EGC ≤ 10 μg/mL, ECG ≤ 5 μg/mL, and EC ≤ 50 μg/mL, the GS cells maintained normal growth (Figure 2A), and there was no significant difference in the cell survival rate between the test groups and the control group (Figure 2B). Therefore, this range was the safe concentration of green tea components.

### 3.2. Preliminary Screening of the Antiviral Effects of Green Tea Components In Vitro

To identify potential antiviral compounds that are effective against SGIV, the antiviral activities of six green tea components were evaluated at their safe working concentrations with a qPCR screening assay in GS cells (Figure 3). The inhibition rate of SGIV infection by each green tea components at its safe working concentration (GTAE at 100 μg/mL, TP at 10 μg/mL, EGCG at 12 μg/mL, EGC at 10 μg/mL, ECG at 5 μg/mL, and EC at 50 μg/mL) was evaluated based on the expression of the *MCP* gene (Figure 3). As shown in Figure 3, there were significant differences in the only SGIV group *MCP* expression with different treatments of each green tea component. The relative antiviral capacities of the green tea components were TP > EGCG > GTAE > ECG > EGC > EC, and the inhibition rates were 99.34%, 98.31%, 98.23%, 88.62%, 73.80%, and 44.31%, respectively. In conclusion, all six green tea components exerted good antiviral effects.

### 3.3. Antiviral Activity of Green Tea Components In Vitro

The antiviral activity of each green tea component (GTAE, TP, and EGCG) against SGIV infection was analyzed in GS cells with light microscopy (Figure 4), Q5c-AFMP (Figure 5), and RT qPCR (Figure 6). The concentrations of each green tea component used in the antiviral activity analysis were GTAE at 100, 50, and 25 μg/mL; TP at 10, 5, and 2.5 μg/mL; and EGCG at 12, 6, and 3 μg/mL.

#### 3.3.1. Antiviral Activity Analysis with Light Microscopy

In parallel with their inhibition efficiencies, the anti-SGIV activities of the green tea components were assessed in more detail in a drug-concentration experiment. As shown in Figure 4, large numbers of typical cytopathic effects (CPEs) appeared in the positive control (SGIV-infected GS cells) after 24 h compared with the control group of normal (uninfected) GS cells, which showed no CPEs. Unlike at 24 h, most SGIV-infected cells no longer attached at 48 h. However, GS cells incubated with SGIV and diluted green tea components (GTAE at 100, 50, or 25 μg/mL; TP at 10, 5, or 2.5 μg/mL; EGCG at 12, 6, or 3 μg/mL) showed no typical CPEs at the safe working concentrations (GTAE at 100 μg/mL, TP at 10 μg/mL, and EGCG at 12 μg/mL), and, as the concentration of each component decreased, CPE increased significantly at 48 h.

#### 3.3.2. Antiviral Activity Analysis with qPCR

The cells were harvested in each group at 48 hpi for an RT qPCR analysis. SGIV *MCP* and *VP19* expression was used to assess their antiviral effects. According to the results shown in Figure 5, there were significant differences in *MCP* expression between cells treated with SGIV and those treated with SGIV and diluted green tea components (GTAE at 100, 50, or 25 μg/mL; TP at 10, 5, or 2.5 μg/mL; and EGCG at 12, 6, or 3 μg/mL). The green tea components dose-dependently reduced *MCP* expression in the SGIV-infected GS cells, with significant differences. *VP19* is an envelope protein, and the results for *VP19* expression were the same as for *MCP* expression. The antiviral effects were greatest at the safe working concentrations of the green tea components and were dose dependent. This experiment revealed that these green tea components (GTAE, TP, and EGCG) significantly and dose-dependently inhibit SGIV infection.

#### 3.3.3. Antiviral Activity Analysis with Q5c-AFMP

Q5c-AFMP detects SGIV-infected host cells, as described previously [2], and was used to monitor the SGIV infection process. As shown in Figure 6, the fluorescence intensity of normal Q5c-AFMP-treated GS cells group was negligible compared with that of the SGIV-infected group, indicating that Q5c specifically binds to SGIV-infected cells. The fluorescence intensity of the target cells incubated with SGIV and green tea components (GTAE, TP, or EGCG) was clearly reduced relative to that of cells incubated with SGIV alone at 48 h. As the green tea component concentrations decreased, the fluorescence intensity increased significantly. These data indicate that these green tea components (GTAE, TP, and EGCG) exert dose-dependent antiviral activities against SGIV infection.

### 3.4. Percentage Inhibition of SGIV Infection by Each Green Tea Component

Based on the antiviral activities determined with RT qPCR (Figure 5), the percentage inhibition of SGIV infection by each green tea component at its safe working concentration (GTAE at 100 μg/mL, TP at 10 μg/mL, and EGCG at 12 μg/mL) was calculated with the analysis of *MCP* and *VP19* expression. The results showed that the percentage inhibition by each green tea component was 99.44% (GTAE), 99.79% (TP), and 99.99% (EGCG) when measured as *MCP* expression, and 98.60% (GTAE), 99.83% (TP), and 99.99% (EGCG) when measured as *VP19* expression. These results show that the inhibition of *VP19* expression was consistent with the inhibition of *MCP* expression (Figure 7). Taken together, these data indicate that GTAE, TP, and EGCG displayed the most effective antiviral activities against SGIV infection.

### 3.5. Antiviral Activity of TP against SGIV In Vivo

The anti-SGIV activity of TP was assessed in vivo. Hybrid groupers were injected intraperitoneally with SGIV mixed with TP at its safe working concentration of 10 μg/mL (SGIV + TP group). The negative and positive control groups were injected intraperitoneally with diluted DMSO and SGIV, respectively. Mortality was recorded for ten days. Spleen tissues collected at 24, 48, and 72 hpi were used for the antiviral analysis. As shown in Figure 8, compared with the SGIV-injection-only group, the expression of *MCP* and *VP19* was significantly reduced in the SGIV + TP group at 24, 48, and 72 hpi (Figure 8). Furthermore, as shown in Figure 9, in the group of fish treated with SGIV only, 100% mortality was observed after 7 days. By comparison, the cumulative mortality in the TP only group and the TP + SGIV group was 8.33% and 11.67% at 10 days, respectively. The survival rate of the hybrid groupers co-injected with TP and SGIV was significantly higher than that of the groupers injected with only SGIV. These data indicate that TP exerts a protective effect against SGIV infection in vivo.

## 4. Discussion

Several green tea components exert good inhibitory effects on SGIV, but their toxicity must be eliminated before use [19]. Cell lines can be used to study the toxicity, immunology, pathogenesis, and biotechnological applications of natural components [20,21]. In the present study, the safe working concentrations of green tea components were first determined in cells. The CCK-8 test results for each green tea component showed safe concentrations of GTAE ≤ 100 μg/mL, TP ≤ 10 μg/mL, EGCG ≤ 12 μg/mL, EGC ≤ 10 μg/mL, ECG ≤ 5 μg/mL, and EC ≤ 50 μg/mL. These data indicated that high concentrations of these components are toxic to cells. The cell viability results for each green tea component were consistent with observations made with light microscopy, confirming that green tea components are not toxic to GS cells at their safe working concentrations.

The antiviral properties of green tea are well known and are mainly attributed to the ability of polyphenols to act as antioxidants and antivirals, to damage cellular membranes, and to prevent the binding and invasion of viruses to cells [22]. These properties are important for the inhibition of virus infections by green tea. Based on the safe concentrations of green tea components for cells, we preliminarily screened several components for antiviral effects. Cell lines play an important role in the early stages of drug screening. Light microscopic observation was used to analyze the antiviral effects of TP, EGCG, and GTAE, and confirmed that normal GS cells were closely linked and undamaged, whereas SGIV-infected cells showed larger numbers of typical CPEs, which is consistent with previous studies [16,18]. GS cells treated with SGIV and GTAE at 100 μg/mL, TP at 10 μg/mL, or EGCG at 12 μg/mL showed no typical CPEs, indicating that GTAE, TP, and EGCG exert good antiviral effects. When the concentration decreased (GTAE at 50 or 25 μg/mL, TP at 5 or 2.5 μg/mL, and EGCG at 6 or 3 μg/mL), the cell morphology changed significantly after SGIV infection. Therefore, the inhibitory effects of GTAE, TP, and EGCG on SGIV infection are concentration dependent. Although light microscopy only detected the presence of an antiviral effect, the RT qPCR results detected the percentage inhibition of SGIV infection afforded by each component. The activities of GTAE, TP, and EGCG against SGIV infection were the greatest of the components tested (all > 98%), indicating that some green tea components have very strong anti-SGIV effects. The relative antiviral capacities of the green tea components were TP > EGCG > GTAE > ECG > EGC > EC, with inhibition rates of 99.34%, 98.31%, 98.23%, 88.62%, 73.80%, and 44.31%, respectively. These results are the same as the antiviral effects of green tea catechins against influenza virus, which showed that EGCG and ECG have better inhibitory activity against the influenza virus than EGC [23].

For accuracy and consistency, the antiviral effects of the green tea components were also analyzed with Q5c-AFMP. Aptamer Q5c, which targets SGIV-infected cells, is a synthetic oligonucleotide selected with the systematic evolution of ligands with exponential enrichment technology in a previous study [2]. Aptamer Q5c recognizes SGIV-infected cells with high specificity and can be used to monitor SGIV infection [8,24]. The results of the Q5c-AFMP analysis were consistent with those of RT qPCR. The green tea components GTAE, TP, and EGCG exerted good antiviral effects and displayed dose-dependent antiviral activities against SGIV. *Illicium verum* Hook. f. extracts also have effective concentration-dependent anti-SGIV properties [16]. Quercetin has been clearly shown to damage SGIV particles and to disturb SGIV binding, invasion, and replication in host cells, with inhibition rates of 76.14%, 56.03%, and 52.73%, respectively [25]. Similarly, components of *Lonicera japonica* Thunb. showed dose-dependent anti-SGIV effects, and *Lonicera japonica* Thunb. aqueous extracts, isochlorogenic acid A, isochlorogenic acid B, isochlorogenic acid C, caffeic acid, and luteolin had the best antiviral activities against SGIV, with percentage inhibitions of >90% [18]. All these findings indicate that some medicinal plants exert dose-dependent effects against SGIV infection. The antiviral effects of green tea components were consistent with previous research. EGCG prevents influenza virus infection by combining with virus hemagglutinin and preventing viral attachment to cell-surface receptors [26]. EGCG also inhibits human immunodeficiency virus type reverse transcription [27]. EC, ECG, and EGCG have been shown to inhibit HIVRT, and the mechanism of action may be the competitive inhibition of template–primer binding or the noncompetitive inhibition of deoxythymidine triphosphate [27,28]. Tea polyphenols have been shown to enhance the body’s immunity to combat COVID-19 infection and other viral infectious diseases [29]. Various mechanisms may underlie the activity of TPs against COVID-19 infection, including the activation of transcription factors, the blocking of cellular receptors, and the inhibition of multiple viral targets [29]. The precise mechanisms by which green tea components inhibit SGIV infection must be explored in future research.

Before effective drugs are widely used in aquaculture, their antiviral activities must be analyzed in vivo. Green tea derivatives have positive effects on the growth, immune systems, protection against pathogens, and blood chemistry of different fish species [30,31,32,33]. In the present study, the anti-SGIV activity of TP was assessed in vivo based on its rate of inhibiting SGIV infection. We recorded the cumulative mortality of infected grouper across ten days. The mortality of fish in the group injected with SGIV only was 100% mortality after 7 days. By comparison, the cumulative mortality in the TP + SGIV group was only 11.67% after 7 days. These results confirm that TP has excellent anti-SGIV effects both in vitro and in vivo, which is consistent with the effects of green tea derivatives against GCRV [14]. However, TP was slightly toxic to fish, so reducing the cytotoxicity of TP and improving the drug utilization rate will be the foci of future research.

## 5. Conclusions

Green tea aqueous extract, tea polyphenols, epigallocatechin-3-gallate, epicatechin gallate, and epigallocatechin displayed excellent antiviral activities against SGIV infection. Tea polyphenols also showed excellent antiviral effects in vivo. These data indicate that some green tea components have great potential utility for the development of an effective treatment against serious SGIV infections in grouper aquaculture.

## Figures and Tables

**Figure 1 viruses-14-01227-f001:**
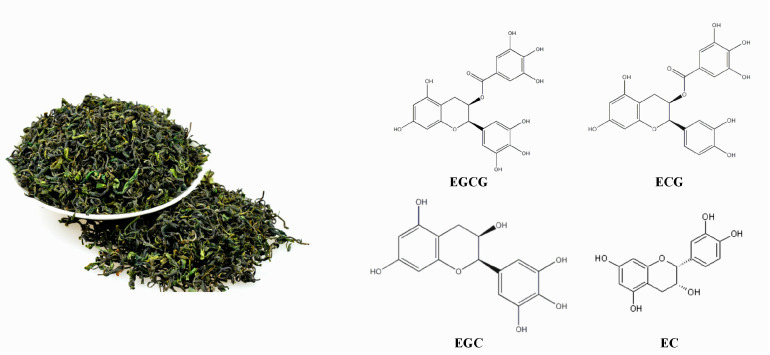
Green tea and structures of major components (EGCG, ECG, EGC, and EG).

**Figure 2 viruses-14-01227-f002:**
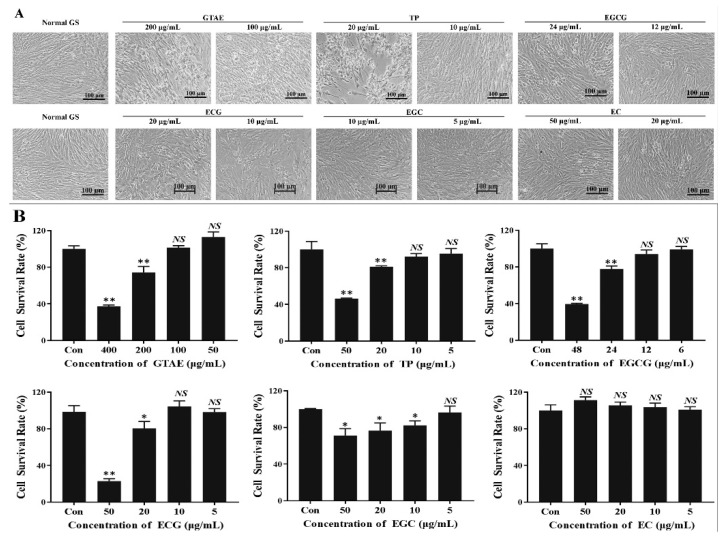
Determining the working concentrations of green tea components. (**A**) Grouper spleen (GS) cells were incubated with different concentrations of green tea components for 48 h and then observed with light microscopy. Morphological changes were observed in the GS cells incubated with green tea aqueous extract (GTAE), TP (tea polyphenols), epigallocatechin-3-gallate (EGCG), epicatechin gallate (ECG), epigallocatechin (EGC), and epicatechin (EC). (**B**) Cell survival analysis with CCK-8 solution: GS cells were incubated with different concentrations of green tea components. The results of comparisons with *p* < 0. 05 were considered to represent statistically significant differences (* *p* < 0. 05, ** *p* < 0. 01, *NS* is no significant difference.). The same below.

**Figure 3 viruses-14-01227-f003:**
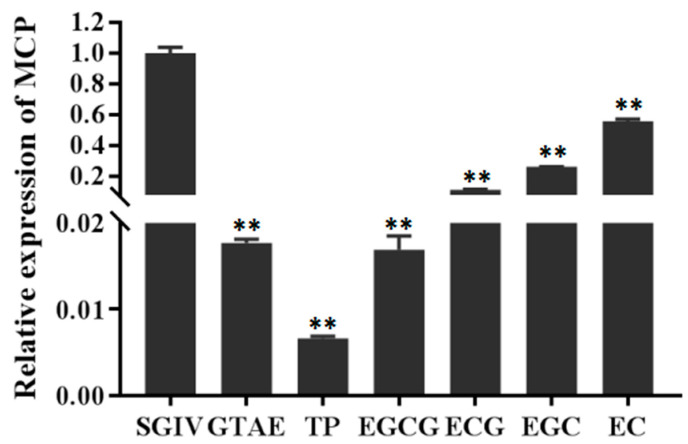
RT qPCR was used to analyze the antiviral effects of green tea components. ** indicates *p* < 0.01.

**Figure 4 viruses-14-01227-f004:**
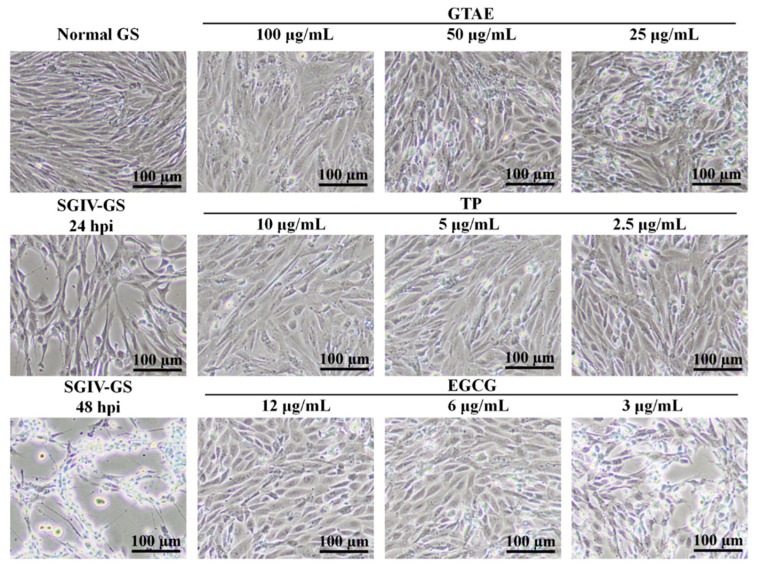
The antiviral activities of green tea components against SGIV observed with light microscopy at 48 h. GS cells incubated with SGIV and diluted green tea components (GTAE at 100, 50, or 25 μg/mL; TP at 10, 5, or 2.5 μg/mL; and EGCG at 12, 6, or 3 μg/mL) constituted the test groups. GS cells infected with SGIV-only constituted the positive control group. GS cells incubated with L15 medium only constituted the negative control group. Each green tea component had the greatest antiviral effect at its safe working concentration (GTAE at 100 μg/mL, TP at 10 μg/mL, and EGCG at 12 μg/mL).

**Figure 5 viruses-14-01227-f005:**
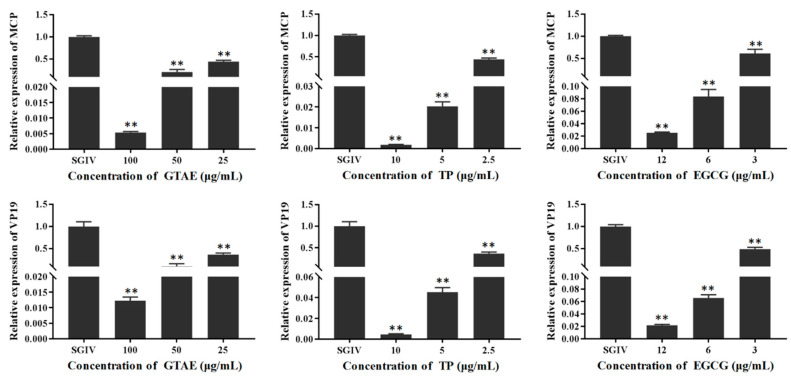
Antiviral activities of green tea components against SGIV tested with qPCR at 48 h. GS cells incubated with SGIV and diluted green tea components (GTAE at 100, 50, or 25 μg/mL; TP at 10, 5, or 2.5 μg/mL; and EGCG at 12, 6, or 3 μg/mL) constituted the test groups. Untreated SGIV-infected GS cells constituted the positive control group. The antiviral effects were analyzed as the inhibition of *MCP* and *VP19* expression. ** indicates *p* < 0.01.

**Figure 6 viruses-14-01227-f006:**
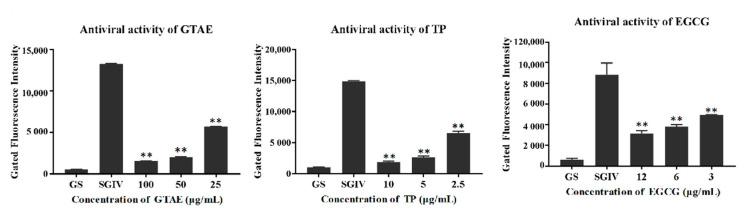
Antiviral activities of green tea components against SGIV evaluated with Q5c-AFMP at 48 h. GS cells incubated with SGIV and the diluted green tea components (GTAE at 100, 50, or 25 μg/mL; TP at 10, 5, or 2.5 μg/mL; and EGCG at 12, 6, or 3 μg/mL) constituted the test groups. GS cells infected with SGIV only constituted the positive control group. Antiviral effects were analyzed as the fluorescence intensity of cells. ** indicates *p* < 0.01.

**Figure 7 viruses-14-01227-f007:**
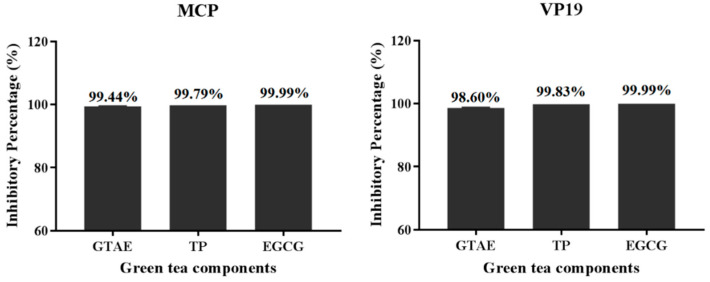
Percentage inhibition analysis of each green tea component against SGIV infection. The percentage inhibition of SGIV infection by each green tea component was 99.44% (GTAE), 99.79% (TP), and 99.99% (EGCG) when measured as *MCP* expression and 98.60% (GTAE), 99.83% (TP), and 99.99% (EGCG) when measured as *VP19* gene expression.

**Figure 8 viruses-14-01227-f008:**
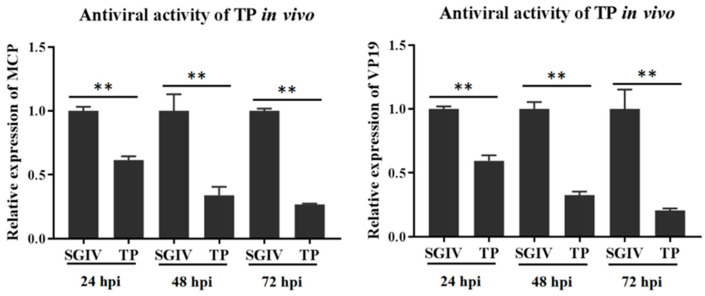
Antiviral activity of green tea component TP against SGIV in vivo. Hybrid groupers were injected intraperitoneally with SGIV mixed with TP at its safe working concentration of 10 μg/mL. Spleen tissues were analyzed for antiviral activity at 24, 48, and 72 hpi. ** indicates *p* < 0.01.

**Figure 9 viruses-14-01227-f009:**
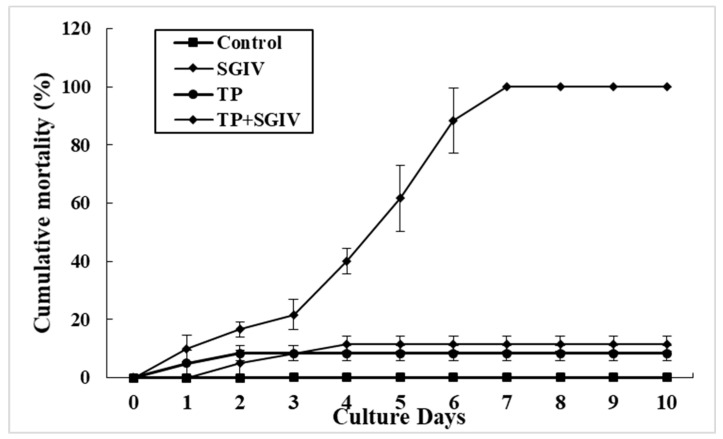
Cumulative mortality over 10 days.

**Table 1 viruses-14-01227-t001:** Primers used to detect SGIV infection with qPCR.

Primer	Sequences
qMCP-F [1]	5′-GCACGCTTCTCTCACCTTCA-3’
qMCP-R	5′-AACGGCAACGGGAGCACTA-3′
qVP19-F [1]	5′-TCCAAGGGAGAAACTGTAAG-3′
qVP19-R	5′-GGGGTAAGCGTGAAGACT-3′
β-actin-F [17]	5′-TACGAGCTGCCTGACGGACA-3′
β-actin-R	5′-GGCTGTGATCTCCTTCTGCA-3′

## Data Availability

The data that support the findings of this study are available from the corresponding author upon reasonable request.

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
