# Peer review of "Antiviral Activities of Green Tea Components against Grouper Iridovirus Infection In Vitro and In Vivo"

_viruses, 2022, doi:10.3390/v14061227_

Round 1
Reviewer 1 Report
The authors examined the effects of Green tea components on Singapore Grouper iridovirus infection and growth in vitro and in vivo. Their results suggests that green tea polyphenols are effective antiviral agents against SGIV under the test conditions provided in this paper.
Comment: In Figure 2. Please define NS in the legend.
Comment: In Figure 4. Please indicate in the legend or Paragraph 3.3.1 at what time after infection and green tea addition were the cell examined in the study. You show the control infected cells after 24 and 48 hours, but you do not say when the photomicrographs were taken, e.g. 24 or 48 hours postinfection. In Paragraph 3.3.2, you harvest the cells at 48 hpi, so the reader would assume the photomicrographs were taken at 48 hpi. However, please state this in Paragraph 3.3.1 and the figure legend.
Comment: In Figure 6, please indicate when (hpi) the samples were taken.
Comment: Although it is a common calculation, the authors should indicate how they made the calculation of percent inhibition.
Conclusion: The authors did demonstrate that green tea components have the potential for being used as antiviral compounds in grouper aquaculture. However, the practical use of green tea polyphenols must be demonstrated, i.e. feeding of green tea components to groupers and then exposure to SGIV, administering green tea components to groupers during an active infection process to see if it will stave off grouper die-offs, etc. I look forward to their next paper.
Author Response
1.Comment: In Figure 2. Please define NS in the legend.
Answer:Thanks for your recommendation. It has been revised in the manuscript (Page 5).
2.Comment: In Figure 4. Please indicate in the legend or Paragraph 3.3.1 at what time after infection and green tea addition were the cell examined in the study. You show the control infected cells after 24 and 48 hours, but you do not say when the photomicrographs were taken, e.g. 24 or 48 hours postinfection. In Paragraph 3.3.2, you harvest the cells at 48 hpi, so the reader would assume the photomicrographs were taken at 48 hpi. However, please state this in Paragraph 3.3.1 and the figure legend.
Answer:Thanks for your recommendation. It has been revised in the manuscript (Page 6).
3.Comment: In Figure 6, please indicate when (hpi) the samples were taken.
Answer:Thanks for your recommendation. It has been revised in the manuscript (Page 8).
4.Comment: Although it is a common calculation, the authors should indicate how they made the calculation of percent inhibition.
Answer:Thanks for your recommendation. It has been revised in the manuscript (Page 4) and added to 2.6.
Conclusion: The authors did demonstrate that green tea components have the potential for being used as antiviral compounds in grouper aquaculture. However, the practical use of green tea polyphenols must be demonstrated, i.e. feeding of green tea components to groupers and then exposure to SGIV, administering green tea components to groupers during an active infection process to see if it will stave off grouper die-offs, etc. I look forward to their next paper.
Thank you for your recommendation! I will take this suggestion into account in my next plan.

Reviewer 2 Report
This paper is devoted to assessing the possibility and effectiveness of using green tea extract and its individual components as an antiviral agent to prevent and treat of SGIV infections in the grouper aquaculture industry. As is known, SHIV can cause mass mortality of fish in a short period, which will lead to serious economic losses in sea bass aquaculture. Given the high pathogenicity of the virus, this work is very relevant. Of particular interest is the use of green tea and its components as an antiviral agent. The authors managed to show a dose-dependent antiviral effect of green tea components against SGIV infection.
Comments and Suggestions for Authors:
- The name of Figure 1 and the decoding of the designations A and B duplicate each other. Probably, it is possible to combine them.
- Why is there no mention of the EC and its concentration on line 179?
- The information given in lines 187-188, 196-197, 202-203 is a repetition.
- As follows from the results shown in Figure 5, GTAE at a concentration of 100 µg/ml inhibits the expression of the VP19 gene much weaker compared to the MCP gene. At the same time, the effectiveness of the TP and EGCG components of GTAE is approximately the same. What is the reason for the observed differences in the action of the extract and its individual components?
- Lines 298-300: “The survival rate of the hybrid groupers co-injected with TP and SHIV was significantly lower than that of the groupers injected with only SHIV”
Probably, after all, the survival rate of hybrid groupers co-injected with TP and SHIV was significantly HIGHER than that of the groupers injected with only SHIV?
- An important question concerning the practical application of green tea components to prevent viral infection of sea bass. How feasible is intraperitoneal administration of antiviral drugs to fish not in the laboratory, but to the fish population in aquaculture?

Author Response
- The name of Figure 1 and the decoding of the designations A and B duplicate each other. Probably, it is possible to combine them.
Answer: Thanks for your recommendation. It has been revised in the manuscript (Page 2). We changed the figure.
- Why is there no mention of the EC and its concentration on line 179?
Answer: Thanks for your recommendation. The highest concentration was used to GS cell, but there was not harmful to GS cells, so it's not mentioned.
- The information given in lines 187-188, 196-197, 202-203 is a repetition.
Answer: Thanks for your recommendation. It has been revised in the manuscript (Page 5). We deleted repeating field.
- As follows from the results shown in Figure 5, GTAE at a concentration of 100 µg/ml inhibits the expression of the VP19 gene much weaker compared to the MCP gene. At the same time, the effectiveness of the TP and EGCG components of GTAE is approximately the same. What is the reason for the observed differences in the action of the extract and its individual components?
Answer : Thanks for your recommendation. MCP is major capsid protein gene and VP19 is an envelope prote.GTAE is green tea extract, and contains a wider variety of chemical component. It is possible that other components except EGCG and TP inhibited MCP expression, but no inhibition of VP19.
- Lines 298-300: “The survival rate of the hybrid groupers co-injected with TP and SHIV was significantly lower than that of the groupers injected with only SHIV”. Probably, after all, the survival rate of hybrid groupers co-injected with TP and SHIV was significantly HIGHER than that of the groupers injected with only SHIV?
Answer: Thanks for your recommendation. It has been revised in the manuscript (Page9).
- An important question concerning the practical application of green tea components to prevent viral infection of sea bass. How feasible is intraperitoneal administration of antiviral drugs to fish not in the laboratory, but to the fish population in aquaculture?
Answer: Thanks for your recommendation. The results of this study are preliminary studies on the antiviral effects of green tea component, our next experiments will base on practical application methods.
Thank you for your advice. We will make improvements in future work. Best wishes!
